# Memory-Efficient Semi-Supervised Continual Learning: The World is its Own Replay Buffer

## Abstract

Rehearsal is a critical component for class-incremental continual learning, yet it requires a substantial memory budget. Our work investigates whether we can significantly reduce this memory budget by leveraging unlabeled data from an agent's environment in a realistic and challenging continual learning paradigm. Specifically, we explore and formalize a novel semi-supervised continual learning (SSCL) setting, where labeled data is scarce yet non-i.i.d. unlabeled data from the agent's environment is plentiful. Importantly, data distributions in the SSCL setting are realistic and therefore reflect object class correlations between, and among, the labeled and unlabeled data distributions. We show that a strategy built on pseudo-labeling, consistency regularization, Out-of-Distribution (OoD) detection, and knowledge distillation reduces forgetting in this setting. Our approach, DistillMatch, increases performance over the state-of-the-art by no less than 8.7% average task accuracy and up to a 54.5% increase in average task accuracy in SSCL CIFAR-100 experiments. Moreover, we demonstrate that DistillMatch can save up to 0.23 stored images per processed unlabeled image compared to the next best method which only saves 0.08. Our results suggest that focusing on realistic correlated distributions is a significantly new perspective, which accentuates the importance of leveraging the world's structure as a continual learning strategy.

## 1 Introduction

Computer vision models in the real-world are often frozen and not updated after deployment, yet they may encounter novel data in the environment. Unlike the typical supervised learning setting, class-incremental continual learning challenges the learner to incorporate new information as it sequentially encounters new object classes without forgetting previously-acquired knowledge (catastrophic forgetting). Research has shown that rehearsal of prior classes is a critical component for class-incremental continual learning (Hsu et al., 2018; van de Ven & Tolias, 2019). Unfortunately, rehearsal requires a substantial memory budget, either in the form of a coreset of stored experiences or a separate learned model to generate samples from past experiences. This is not acceptable for memory-constrained applications which cannot afford to increase the size of their memory as they encounter new classes.

Instead, we consider a novel real-world setting where an incremental learner's labeled task data is a product of its environment and the learner encounters a vast stream of *unlabeled* data in addition to the labeled task data. In such a setting (visualized in Figure 1), the unlabeled datastream is intrinsically correlated to each learning tasks due to the underlying structure of the environment. We explore many ways in which this correlation may exist. For example, when an incremental learner is tasked to learn samples of the previously-unseen class $c_i$ at time $i$ in the real world, examples of $c_i$ may be encountered in the environment (in unlabeled form) during some future task. In such a setting, an incremental learner could use the unlabeled data in its environment as a source of memory-free rehearsal, though it would need a method to determine which unlabeled data is relevant to the incremental task (i.e. detecting in-distribution data).

We formalize this realistic paradigm in the *semi-supervised continual learning* (SSCL) setting, wherein unlabeled and labeled data are not i.i.d. as they are correlated through the underlying structure of the environment. We propose and conduct experiments over a realistic setting in which this correlation may exist, in the form of label super-class structure (e.g. *unlabeled* examples

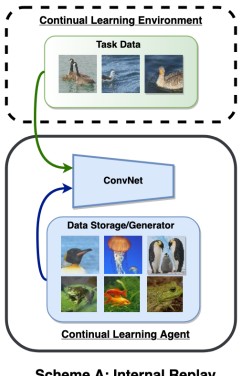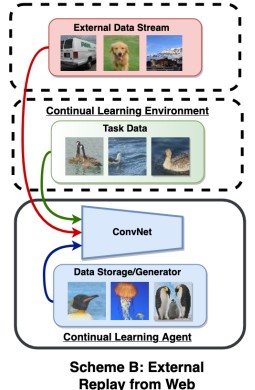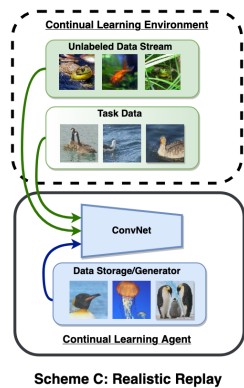

Figure 1: Unlike standard replay (scheme A) which requires a substantial memory budget, we explore the potential of an unlabeled datastream to serve as replay and significantly reduce the required memory budget. Unlike previous work which requires access to an external datastream uncorrelated to the environment (scheme B), we consider the datastream to be a product of the *continual learning agent's environment* (scheme C).

of household furniture such as chairs, couches, and tables will appear while learning the *labeled* examples of household electrical devices such as lamp, keyboard, and television (Krizhevsky et al., 2009; Zhu & Bain, 2017)). We measure the final-task accuracy $A$, the accuracy over all tasks $\Omega$, and the coreset memory required to attain a specific level of $\Omega$ accuracy over several realistic SSCL settings. *Our experiments demonstrate that state-of-the-art continual learning methods (Lee et al., 2019) perform inconsistently in the novel SSCL paradigm with no prior method performing "best" across all settings*. This leads us to ask "How can an approach to catastrophic forgetting be robust to several realistic, memory-constrained continual learning scenarios?"

To answer the above question, we propose a novel learning approach that works well in both the simple (i.e., no correlations) and realistic SSCL settings: DistillMatch. We leverage unlabeled data not only for knowledge distillation (in which the distilling model is fixed), but also for a semi-supervised loss (in which the supervisory signal can adapt during training on the new task). Key to our approach is that we address the distribution mismatch between the labeled and unlabeled data (Oliver et al., 2018) with out-of-distribution (OoD) detection (and are the first to do so in the continual learning setting). Compared to nearest prior state-of-the-art (all methods from (Lee et al., 2019)) configured to work as well as possible in the novel SSCL setting, we outperform the state-of-the-art in all of our experiment scenarios by as much as a **54.5%** increase in $\Omega$ and no less than an **8.7%** increase. Furthermore, we find that our method can save up to **0.23** stored images per processed unlabeled image over naive rehearsal (compared to (Lee et al., 2019) which only saved 0.08 stored images per processed unlabeled image). *In summary, we make the following contributions:*

1. We propose the realistic semi-supervised continual learning (SSCL) setting, where object-object correlations between labeled and unlabeled sets are maintained through a label super-class structure. We show that state-of-the-art continual learning methods perform inconsistently in the SSCL setting (i.e. no baseline method is "best" across all settings).

2. We propose a novel continual learning method *DistillMatch* for the SSCL setting leveraging pseudo-labeling, strong data augmentations, and out-of-distribution detection. Compared to the baselines, *DistillMatch* achieves superior performance on a majority of metrics for **8/8** experiments and results in substantial memory budget savings.

## 2 BACKGROUND AND RELATED WORK

**Knowledge Distillation in Continual Learning**: Several related methods leverage distillation losses on past tasks to mitigate catastrophic forgetting using soft labels from a frozen copy of the previous task's model (Castro et al., 2018; Hou et al., 2018; Li & Hoiem, 2017; Rebuffi et al., 2017). For example, learning using two teachers, with one teacher distilling knowledge from previous tasks and another distilling knowledge from the current task, has been found to increase adaptability to a new task while preserving knowledge on the previous tasks (Hou et al., 2018; Lee et al., 2019). Class-balancing and fine-tuning have been used to encourage the model's final predicted class distribution

to be balanced across all tasks (Castro et al., 2018; Lee et al., 2019). These methods are related in that they rely on distillation losses to mitigate catastrophic forgetting, but the losses are designed to distill knowledge about specific local tasks and cannot discriminate between classes from different tasks (crucial for class incremental learning). More context on where our work fits into the greater body of continual learning research is provided in Appendix H.

Global distillation (GD) introduces a global teacher which provides a knowledge ensemble from both the past tasks and current task (Lee et al., 2019). This addresses a crucial shortcoming of common knowledge distillation methods which do not reconcile information from the local tasks (i.e. the groups of object classes presented sequentially to the learner) with the global task (i.e. all object classes seen at any time). GD leverages a large stream of uncorrelated unlabeled data from sources such as data mining social media or web data (Lee et al., 2019) to boost its distillation performance. Similar to GD, we leverage an unlabeled datastream to mitigate forgetting, but we take the perspective that this datastream is from the agent's environment and reflects object-object correlation structures imposed by the world (i.e. correlations between the task data and the unlabeled data).

**Out of Distribution Detection**: Leveraging unlabeled data for rehearsal is key to our work, but it can contain a mix of classes not in the distribution of the data seen by the learner thus far. Therefore, we include Out-of-Distribution (OoD) Detection (Hsu et al., 2020; Lee et al., 2018; Liang et al., 2017) to select unlabeled data corresponding to the classes our learner has seen so far with high confidence. Semantic OoD detection is a difficult challenge (Hsu et al., 2020) and we do not have access to any known OoD data to calibrate our confidence. We therefore build on a recent method, Decomposed Confidence (DeConf) (Hsu et al., 2020), which can be calibrated using only in-distribution training data. The method consists of decomposed confidence scoring with a learned temperature scaling in addition to input pre-processing. For further details, the reader is referred to (Hsu et al., 2020).

**Semi-Supervised Learning**: In semi-supervised learning (which motivates the SSCL setting), models are given a (typically small) amount of labeled data and leverage unlabeled data to boost performance. This is an active area of research given that large, labeled datasets are expensive, but most applications have access to plentiful, cheap unlabeled data. There are several approaches to semi-supervised learning (Berthelot et al., 2019; Lee, 2013; Kingma et al., 2014; Kuo et al., 2019; Miyato et al., 2018; Oliver et al., 2018; Sohn et al., 2020; Springenberg, 2015; Tarvainen & Valpola, 2017) which involve balancing a supervised loss $\ell_s$ applied to the labeled data with an unsupervised loss $\ell_{ul}$ applied to unlabeled data. Additional details on these methods are provided in Appendix H.

## 3   SSCL SETTING

In class-incremental continual learning, a model is gradually introduced to labeled data corresponding to $M$ semantic object classes $c_1, c_2, \ldots, c_M$ over a series of $N$ tasks, where tasks are non-overlapping subsets of classes. We use the notation $\mathcal{T}_n$ to denote the set of classes introduced in task $n$, with $|\mathcal{T}_n|$ denoting the number of object classes in task $n$. Each class appears in only a single task, and the goal is to incrementally learn to classify new object classes as they are introduced while retaining performance on previously learned classes. The class-incremental learning setting (Hsu et al. (2018)) is a challenging continual learning settings because no task indexes are provided to the learner during inference and the learner must support classification across all classes seen up to task $n$.

We extend the class-incremental continual learning setting in the realistic semi-supervised continual learning (SSCL) setting, where data distributions reflect existing object class correlations between, and among, the labeled and unlabeled data distributions. The amount of labeled data in this setting is drastically reduced as is common in semi-supervised learning. For example, our experiments reduce the number of labeled examples per class by 80% compared to a prior setting (Lee et al., 2019). At task $n$, we denote batches of labeled data as $\mathcal{X}_n = \{(x_b, y_b) \colon b \in (1, \cdots, B) \mid y_b \in \mathcal{T}_n\}$ and batches of unlabeled data as $\mathcal{U}_n = \{u_b \colon b \in (1, \cdots, \mu B)\}$. Here, $B$ refers to batch-size and $\mu$ is a hyperparameter describing the relative size of $\mathcal{X}_n$ to $\mathcal{U}_n$. The goal in task $n$ is to learn a model $\theta_n$ which predicts object class labels for any query input over all classes seen in the current and previous tasks ($\mathcal{T}_1 \cup \mathcal{T}_2 \cup \cdots \cup \mathcal{T}_n$). The index $n$ on $\theta_n$ indicates that our model is updated each task; i.e. $\theta_{n-1}$ refers to model from the previous task and $\theta_n$ refers to the model from the current task.

To simulate an environment where unlabeled and labeled data are naturally correlated, we leverage well-defined relationships between objects derived from a *super-class* structure (i.e. various animals

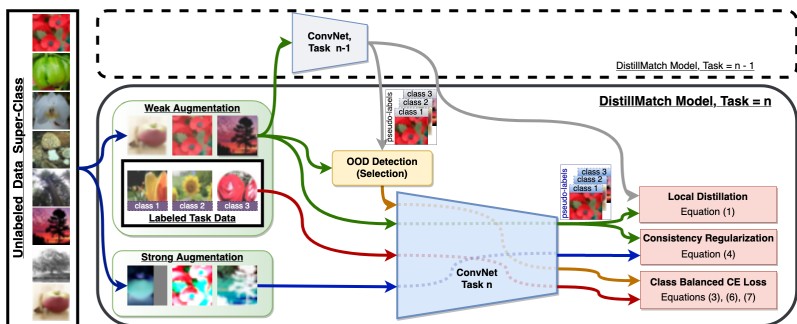

Figure 2: Diagram of DistillMatch, our proposed algorithm for semi-supervised continual learning. We use unlabeled data from the agent's environment to ground the current model in the past task while simultaneously learning new knowledge. This is done with a combination of local distillation, consistency regularization, and pseudo-labeling unlabeled data with confident predictions from an Out-of-Distribution Detector.

within one super-class). We use the CIFAR-100 dataset (Krizhevsky et al., 2009) because object correlations among classes and parent classes, crucial to our experiments, are well defined and explored (Zhu & Bain, 2017). This dataset contains eight unbalanced super-classes, which we use to simulate realistic data environments. Each super-class contains a number of parent classes (e.g. one super-class contains the parent classes flowers, fruits/vegetables, and trees). There are 20 parent classes in total which form the 20 continual learning tasks, with each parent class consisting of five object classes (e.g. flowers parent class consists of orchids, poppies, roses, sunflowers, tulips). For a single task, when our learner is being shown labeled training data from one of the parent classes (e.g. flowers, fruit/vegetables, *or* trees), the unlabeled data for this task will contain examples from the entire super-class (e.g. flowers, fruits/vegetables, *and* trees).

SSCL with this realistic super-class "environment" structure is our main setting, but we also explore several other correlation combinations, including the simple SSCL setting without any super-class structure. We use the following terminology to describe the correlations of the tasks (i.e. labeled data): *RandomClass Tasks*, where no correlations exist in task classes, and *ParentClass Tasks*, where tasks are introduced by CIFAR-100 parent classes (i.e. each task is to learn the five classes of a single CIFAR-100 parent class). For the unlabeled data distribution we have: *Uniform Unlabeled*, where all classes are uniformly dsitributed in unlabeled data for all tasks, *PositiveSuperclass Unlabeled*, where the unlabeled data of each tasks consists of the parent classes in the same super-class as the current task, *NegativeSuperclass Unlabeled*, where the unlabeled data of each tasks consists of parent classes from different super-class as the current task, and *RandomUnlabeled*, where the unlabeled data of each task consists of 20 randomly sampled classes (roughly equal to the average class size in a super-class). Further discussion and details, including figures depicting example streams for each task sequence, are provided in Appendix F.

## 4 APPROACH

**Local Distillation - Preserve Past Tasks** $1 \dots n-1$ **Using Unlabeled Data:** Our approach, summarized in Figure 2, is a distillation-based approach and therefore uses a standard local distillation loss (as in prior methods). The intuition of knowledge distillation is that the current model should make similar predictions to previous models over the set of classes associated with the previous tasks. Refining prior notation, this loss depends on $\theta_{i,n}$: the model of $\theta$ at time $i$ that has been trained up to, and including, data with the classes from task $n$. For example, $\theta_{n,1:n}$ refers to the model trained during task $n$ and its logits associated with all tasks up to and including class $n$. Let us denote $p_\theta(y \mid x)$ as the predicted class distribution produced by model $\theta$ for input $x$. Using this notation, $\ell_{dst}$ during the training of task $n$ is given as:

$$\ell_{dst} = \frac{1}{n-1} \sum_{t=1}^{n-1} \left( \frac{1}{B} \sum_{b=1}^{B} \mathcal{L}_{dst}(p_{\theta_{n-1,t}}(y \mid \alpha(x_b)), p_{\theta_{n,t}}(y \mid \alpha(x_b))) \right) \tag{1}$$

where $\mathcal{L}_{dst}$ is a distance loss such as KL divergence, and $\alpha$ denotes weak data augmentations such as random horizontal flips and crops. This loss acts as a regularization penalty to encourage the current model to make similar predictions to the previous model for all tasks $1 \dots n-1$.

**OoD Detector Training and Calibration:** We train and calibrate an OoD detector in the SSCL setting in order to identify unlabeled data previously seen by the agent. OoD detection calculates a scalar score $\mathcal{S}(u_b)$ for unlabeled input $u_b$, and rejects $u_b$ as out-of-distribution if $\mathcal{S}(u_b) < \tau_{OoD}$, where $\tau_{OoD}$ is a calibrated threshold. OoD scores in our SSCL setting include the time index $n$, $\mathcal{S}_n(u_b)$, and are generated from a separate OoD model $\phi_n$ using the method in (Hsu et al., 2020). We use unlabeled data considered as in-distribution for a hard distillation loss, and we use all unlabeled data for soft knowledge distillation and consistency losses. We use a separate model because calibrating the decision threshold $\tau_{OoD}$ requires labeled hold-out data, which we cannot afford to sacrifice in our main classification model given we are already working in a limited labeled data regime. We hold-out 50% of the labeled training data (across all tasks) when training $\phi_n$ for this calibration decision. At the end of each task $n$, we calibrate $\tau_{OoD}$ to operate at a $\delta\%$ true-positive ratio (TPR) using the hold-out labeled data, where $\delta$ is a scalar hyperparameter. For computational efficiency, we exclude unlabeled losses when training $\phi_n$. We note that our implementation uses the same number of models (and therefore parameters) as used in the prior state-of-the-art method GD (Lee et al., 2019).

**Confidence-Based Hard Distillation - Preserve Past Tasks** $1 \ldots n - 1$ **Using Unlabeled Data:** In our experiments, we found that eq. (1) works well at preserving performance on local tasks, but does not work well on distilling knowledge from the global task (i.e. object classes from tasks $1 \ldots n$). We demonstrate hard distillation (cross entropy loss with a one-hot vector label) is preferable over soft distillation because hard distillation distills knowledge across all classes (i.e. the global task). Specifically, we distill global knowledge from task $n - 1$ into task $n$ by using a frozen copy of $\theta_{n-1,1:n-1}$ to pseudo-label unlabeled data by generating predicted class distributions for unlabeled images as: $q_{b,n-1} = p_{\theta_{n-1,1:n-1}}(y \mid \alpha(u_b))$, and one-hot pseudo-labels as $\hat{q}_{b,n-1} = \arg\max(q_{b,n-1})$. We identify highly confident data on task $n - 1$ using the DeConf OoD detection described in Section 2. Let $\mathcal{S}_{n-1}(u_b)$ denote the score of our OoD detector for past task classes of our pseudo-label model, $\theta_{n-1,1:n-1}$. Then, our confidence-based hard distillation loss for the pseudo-labeled data $\ell_{pl}$ becomes:

$$\ell_{pl} = \frac{1}{B_{pl}} \sum_{b=1}^{\mu B} \mathbb{1}(\mathcal{S}_{n-1}(u_b) \geq \tau_{OoD}) \mathcal{L}_{CE}(q_{b,n-1}, p_{\theta_{n,1:n}}(y \mid \alpha(u_b))) \tag{2}$$

where $\mathcal{L}_{CE}(p, q)$ is the cross-entropy between probability distributions $p$ and $q$ and $B_{pl}$ is the number of pseudo-labeled examples in the given batch identified with OoD detection. We normalize the batch by $B_{pl}$ and not $\mu B$ so that $\ell_{pl}$ can be class balanced alongside the labeled training data (described in the next subsection). Note that we only need to retain $\theta_{n-1,1:n-1}$ at task $n$.

**Semi-Supervised Class Balancing - Balance Past Tasks** $1 \ldots n - 1$ **with New Task** $n$ **Using Labeled and Unlabeled Data:** Our method is designed to work even in the absence of a coreset (stored images) when unlabeled data from past tasks is available. This is due to the design choice of including hard pseudo-labeled data from a frozen model copy. A problem arises from this approach in sensitivity to class imbalance, however. Specifically, the distribution of classes for a given batch becomes imbalanced when considering both labels $y$ and pseudo-labels $\hat{q}_{b,n-1}$. This is because the number of examples per class in the labeled training data is unlikely to equal the number of examples per class in the pseudo-labeled training data, given these come from two different distributions. To address this, we weight loss components proportionally to the distribution of both labeled training data and hard pseudo-labeled unlabeled data. Specifically, we scale the gradient computed from a data with label or pseudo-label $k \in (1, \cdots, K)$ by:

$$w(k) = \frac{1}{K} \cdot \left( \frac{|\{(x,y) \in \mathcal{X}_n\}| + |\{u \in \mathcal{U}_n\}|}{|\{(x,y) \in \mathcal{X}_n \mid y = k\}| + |\{u \in \mathcal{U}_n \mid \hat{q}_{n-1} = k\}|} \right) \tag{3}$$

**Consistency Regularization - Learn New Task** $n$ **Using Unlabeled Data:** We examine the effects of a consistency loss introduced in FixMatch (Sohn et al., 2020) in the SSCL setting to leverage unlabeled data for learning the current task (rather than only preserving knowledge on the past tasks). This loss enforces consistency between weakly and strongly augmented versions of unlabeled data which increases robust decisions on highly confident unlabeled examples. Strong data augmentations, denoted as $\mathcal{A}$, include RandAugment (Cubuk et al., 2019). The model generates a predicted class distribution from a weakly-augmented version of the unlabeled image: $q_b = p_{\theta_{n,1:n}}(y \mid \alpha(u_b))$.

Using a generated one-hot pseudo-label $\hat{q}_b = \arg\max(q_b)$, the unlabeled loss $\ell_{ul}$ is calculated as:

$$\ell_{ul} = \frac{1}{\mu B} \sum_{b=1}^{\mu B} \mathbb{1}(\max(q_b) \geq \tau_{FM}) \mathcal{L}_{CE}(\hat{q}_b, p_{\theta_{n,1:n}}(y \mid \mathcal{A}(u_b))) \tag{4}$$

where $\tau_{FM}$ is a confidence threshold (scalar hyperparameter) above which a pseudo-label is retained.

**Final Loss** - Our final, balanced loss $\ell_{total}$ is given as:

$$\ell_{total} = \frac{1}{B + B_{pl}} \cdot (\ell_s + \ell_{pl}) + \lambda_{ucl}\ell_{ul} + \lambda_{dst}\ell_{dst} \tag{5}$$

$$\ell_s = \sum_{b=1}^{B} w(y_b) \cdot \mathcal{L}_{CE}(y_b, p_{\theta_{n,1:n}}(y \mid \alpha(x_b))) \tag{6}$$

$$\ell_{pl} = \sum_{b=1}^{\mu B} w(\hat{q}_{b,n-1}) \cdot \mathbb{1}(\mathcal{S}_{n-1}(u_b) \geq \tau_{OoD}) \mathcal{L}_{CE}(\hat{q}_{b,n-1}, p_{\theta_{n,1:n}}(y \mid \alpha(u_b))) \tag{7}$$

where $\ell_{ul}$ is taken from eq. (4), $\ell_s$ is the supervised task loss, $\ell_{dst}$ is from eq. (1), and $\lambda_{ucl}$, $\lambda_{dst}$ are hyperparameters which weight of unlabeled consistency loss and local distillation, respectively.

## 5 EXPERIMENTS

We evaluate DistillMatch and state-of-the-art baselines under several realistic SSLC scenarios with the CIFAR-100 dataset using 100 labeled examples per class. We choose recent distillation methods which can leverage the unlabeled data with distillation losses: Distillation and Retrospection (DR) (Hou et al., 2018), End-to-End incremental learning (E2E) (Castro et al., 2018), and Global Distillation (GD) (Lee et al., 2019). Similar to GD (Lee et al., 2019), we do not compare to model-regularization methods (i.e., methods which penalize changes to model parameters, as discussed in Appendix H) because distillation methods have been found to perform better in the class-incremental learning setting (Lee et al., 2019). Besides, these methods are orthogonal to our contribution and could be combined with our approach (and the competing approaches) for better performance. We use implementations of DR and E2E from (Lee et al., 2019), which are adapted from incremental task learning to incremental class learning. These implementations of DR and E2E use the unlabeled data for their respective knowledge distillation loss(es), as shown in Appendix J. We also compare to a neural network trained only with cross entropy loss on labeled data (Base).

### 5.1 METRICS

We evaluate our methods using: (I) final performance, or the performance with respect to all past classes after having seen all $N$ tasks (referred to as $A_{N,1:N}$); and (II) $\Omega$, or the average (over all tasks) normalized task accuracy with respect to an offline oracle method (Hayes & Kanan, 2019). As before, we use index $i$ to index tasks through time and index $n$ to index tasks with respect to test/validation data (for example, $A_{i,n}$ describes the accuracy of our model after task $i$ on task $n$ data). Specifically:

$$A_{i,n} = \frac{1}{|\mathcal{D}_n^{test}|} \sum_{(x,y) \in \mathcal{D}_n^{test}} \mathbb{1}(\hat{y}(x, \theta_{i,n}) = y \mid \hat{y} \in \mathcal{T}_n) \tag{8}$$

$$\Omega = \frac{1}{N} \sum_{i=1}^{N} \sum_{n=1}^{i} \frac{|\mathcal{T}_n|}{|\mathcal{T}_{1:i}|} \frac{A_{i,1:n}}{A_{offline,1:n}} \tag{9}$$

$\Omega$ is designed to evaluate the global task and is therefore computed with respect to all previous classes. For the final task accuracy in our results, we will denote $A_{N,1:N}$ as simply $A_N$.

Table 1: Results (%) CIFAR-100 with 20% Labeled Data for (a) RandomClass Tasks on for various task sizes (no Coreset, Uniform Unlabeled Data Distribution) and (b) ParentClass Task for various coreset sizes (20 Tasks, Uniform Unlabeled Data Distribution). Results are reported as an average of 3 runs, with std provided in the supplementary material. UB refers to the upper bound, given by the offline oracle.

| (a) | | | | | | | (b) | | | | |
|---|---|---|---|---|---|---|---|---|---|---|---|
| Tasks | 5 | | 10 | | 20 | | Coreset | 0 | | 400 | |
| Metric (↑) | $A_N$ | $\Omega$ | $A_N$ | $\Omega$ | $A_N$ | $\Omega$ | Metric (↑) | $A_N$ | $\Omega$ | $A_N$ | $\Omega$ |
| UB | 56.7 | 100 | 56.7 | 100 | 56.7 | 100 | UB | 56.7 | 100 | 56.7 | 100 |
| Base | 15.6 | 52.5 | 8.2 | 34.7 | 4.3 | 22.0 | Base | 3.5 | 18.5 | 14.6 | 53.4 |
| E2E | 12.5 | 46.1 | 7.5 | 32.3 | 4.0 | 21.1 | E2E | 3.2 | 18.1 | 19.5 | 59.3 |
| DR | 16.0 | 53.7 | 8.3 | 36.4 | 4.3 | 22.4 | DR | 3.7 | 19.4 | 20.1 | 57.8 |
| GD | 32.1 | 69.9 | 21.4 | 60.0 | 13.4 | 42.7 | GD | 10.5 | 37.4 | 21.4 | 57.7 |
| DM | **44.8** | **84.4** | **37.5** | **76.9** | **21.0** | **60.8** | DM | **20.8** | **57.8** | **24.4** | **67.5** |

## 5.2 OTHER DETAILS

We do not tune hyperparameters on the full task set because tuning hyperparameters with hold out data from all tasks may violate the principal of continual learning that states each task in visited only once (van de Ven & Tolias, 2019). Following (Lee et al., 2019), we include a coreset for many experiments (although we show our method does not need a coreset when certain object correlations are present between unlabeled and labeled data) which is used for both the labeled cross entropy loss and the distillation loss. Given that our SSL tasks use 20% of the labeled data from (Lee et al., 2019), we also reduce the coreset to 20% of their coreset size (from 2000 to 400). Notice that no methods have a *memory* budget for unlabeled data (i.e. the unlabeled data is from a stream and *discarded after use* rather than stored). We include supplementary details and metrics in our Appendix: additional ablations (A), additional experiment details (B), hyperparameter selection (C), full results including standard deviation and plots (D), OoD performance (E), and super-parent class associations (F).

## 5.3 RESULTS

**We do not need to store coreset data in an environment with uniform unlabeled data.** We first evaluate our methods in a scenario where unlabeled data is drawn from a uniform distribution over all classes (Table 1a). This represents the situation where labeled task data is presented sequentially by a teacher but unlabeled data from all tasks is freely available. We evaluate these experiments with no coreset for replay given that unlabeled examples for all classes are present in each task. Across these experiments, we see that our proposed method (DM) establishes strong performance for SSCL. With respect to final classification accuracy $A_N$ and average normalized task accuracy $\Omega$, we see that DM has considerably higher performance. The local distillation baselines E2E and DR perform no better than Base, which makes sense because local distillation does not distill the global task. The results suggest that when the unlabeled data distribution contains all past task classes, methods which distill across the global task (GD and DM) achieve high performance despite having no coreset.

**Class correlations in the labeled task data negatively affect performance of all distillation methods.** We explore a scenario where object correlations exist in the class distribution of the *labeled* data but not the unlabeled data (Table 1b). This extends the previous scenario by considering learning tasks of similar object types. We evaluate these experiments with and without a coreset to make direct comparisons with both the previous experiment and the next experiment. Compared to the previous experiments, we see that the introduction of super-classes entails a more difficult problem, as every method decreases in performance; yet DM largely outperforms all other methods. For example, in Table 1b) (no coreset), GD results in $37.4\%$ $\Omega$ and DM results in $57.8\%$ $\Omega$, indicating a $54.5\%$ increase over state of the art. Overall, these results pose an interesting question of why parent class tasks would negatively affect performance; we plan to more fully explore this in future works.

**The DistillMatch approach is robust to several *realistic* unlabeled data correlations; the other methods are not.** We evaluate our methods with several unlabeled data distributions (beyond a uniform distribution) with the above ParentClass Tasks (Table 2a). The unlabeled data correlations represent the *realistic SSCL scenario*, a vastly different perspective compared to prior experimental methodologies. We evaluate these experiments with a coreset of 400 images for replay and show that **our method is state of art in each realistic scenario.** Surprisingly, we found that the global

Table 2: Results (%) CIFAR-100 with 20% Labeled Data for (a) ParentClass Task on CIFAR-100 with 20% Labeled Data for various unlabeled data distributions (20 Tasks, with 400 image Coreset) and (b) Selected Ablation Studies (RandomClass Tasks with Uniform Unlabeled Data Distribution, 10 Tasks, no Coreset). Results are reported as an average of 3 runs, with std provided in the supplementary material. For (b), Each row represents a part of our method which is removed as part of the study. UB refers to the upper bound, given by the offline oracle.

(a)

| UL Data Corr. | Positive | | Negative | | Random Sample | |
|---|---|---|---|---|---|---|
| Metric ($\uparrow$) | $A_N$ | $\Omega$ | $A_N$ | $\Omega$ | $A_N$ | $\Omega$ |
| UB | 56.7 | 100 | 56.7 | 100 | 56.7 | 100 |
| Base | 14.6 | 53.4 | 14.6 | 53.4 | 14.6 | 53.4 |
| E2E | 18.9 | 59.4 | 19.9 | 60.1 | 19.8 | 60.0 |
| DR | 18.8 | 62.8 | 20.1 | 62.1 | 19.9 | 61.8 |
| GD | 17.9 | 50.2 | 18.1 | 50.5 | 21.3 | 59.9 |
| DM | **19.7** | **63.3** | **20.7** | **64.8** | **22.4** | **65.1** |

(b)

| Ablation | $A_N$ | $\Omega$ |
|---|---|---|
| $\ell_{pl}$ - eq. (7) | 7.7 | 32.0 |
| $w(k)$ - eq. (3) | 30.2 | 69.6 |
| $\ell_{ul}$ - eq. (4) | 33.3 | 71.2 |
| $\ell_{dst}$ - eq. (1) | 35.2 | 74.1 |
| Full Method | 37.5 | 76.9 |

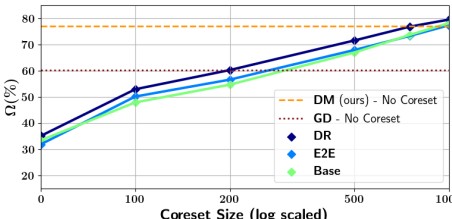

Figure 3: $\Omega$ (%) vs Coreset Size for Base, with GD and DM (no corset) plotted horizontally. We show base requires roughly **935** and **338** stored images to match the performance of DM and GD, respectfully. This is equivalent to **0.23** and **0.08** stored images per unlabeled image (calculated using the number of unlabeled images per task in this scenario, which is 4,000).

distillation method actually performs worse than Base. This can be explained in that the global distillation loss expects the unlabeled data to contain examples which are either in the distribution of the past tasks or out of distribution for the global task (i.e. it assumes the unlabeled data does not contain examples of the current task), and actually hurts performance when this assumption is not held. Our method, on the other hand, performs the same or better compared to the other methods (DR, E2E) which have no such an assumption. These results suggest that our reminding based solution is the only evaluated method which performs well in a memory constrained setting regardless of the correlations between unlabeled and labeled data.

**The DistillMatch approach saves memory.** Here, we perform a study to quantify the memory budget savings from our method (Figure 3). We see that DM performs considerably higher than the leading baseline, GD, in the RandomClass Tasks with Uniform Unlabeled Data Distribution scenario (10 task). All other methods require a large coreset to match DM. We specifically find the necessary coreset size for Base required to match our performance. We show base requires roughly 935 and 338 stored images to match the performance of DM and GD, respectfully. This is equivalent to 0.23 and 0.08 stored images per unlabeled image (calculated using the number of unlabeled images per task in this scenario, which is 4000).

**Each component of DistillMatch contributes to its performance gains.** We perform an extensive ablation study of our method in Table 2b for the scenario with no coreset (further ablations are found in Appendix-A). We show that while the semi-supervised consistency loss (4), class balancing (3), and soft distillation loss (1) are all important to our method, the most important contribution is the hard distillation loss (7). In summary, we find that our design performs well *across a range of conditions*, namely with different amounts of coresets (including none at all), as well as under different unlabeled distributions. This is a key aspect of our experimental setting, which better reflects the flexibility that is necessary from a continual learning algorithm.

## 6 CONCLUSIONS

We formalize the SSCL setting, which mirrors the underlying structure in the real world where a continual agent's learning task is a product of its environment, and determine its effect on the continual learning problem. To address the unique challenges of the SSCL setting, we propose a

novel learning approach that works within these constraints, DistillMatch, notably outperforming closest prior art. Our approach consists of pseudo-labeling and consistency regularization, distillation for continual learning, and out-of-distribution (OoD) detection for confidence selection. Our analysis shows that our reminding-based approach performs well in a memory constrained setting regardless of the correlations between unlabeled and labeled data, unlike existing approaches. A challenge for future work is increasing the effectiveness of semantic OoD detection, and exploring better techniques for calibrating OoD detectors in a continual (online) manner. We acknowledge other concurrent works push realistic continual learning settings as well (Mundt et al., 2020; Ren et al., 2020), but the contributions are orthogonal to our setting.

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
