# OpenReview forum: "Memory-Efficient Semi-Supervised Continual Learning: The World is its Own Replay Buffer"
_ICLR.cc/2021/Conference — Reject_

### Official Review · AnonReviewer1 · 2020-10-25
**Proposes a semisupervised task scenario for continual learning**

**Rating:** 4
**Confidence:** 5

**Review:**

This paper comes up with a novel scenario where the unlabled data are available as well as labeled data in the continual learning scenario.

### Overall
- Based on my understanding, the major contribution is the proposal of a task scenario, aka, experimental setting. The novelty of DistillMatch is an incremental modification of previous work.
- The task setting sidesteps the learning with non-stationarity problem than solving it.
- Further, this setting potentially makes the task easier for the proposed method. To verify whether this is true, more information are needed.
- The presentation of the paper needs polishing, I listed a few points below.

### Pros
- The novel scenario of semisupervised continual learning is proposed. The argument is that in several realistic scenarios old data are often re-observed without label (The funiture labeling example). Therefore instead of storing a coreset, one may make use of the unlabeled data for pseudo-rehearsal/distillation. It is reasonable to make use of it when this assumption is true.
- With the setting the author proposed, the DistillMatch method is able to perform better than previous methods.

### Cons
1. The novelty mostly comes from the task scenario, the DistillMatch method is incremental.
2. Although SSCL is a new scenario, and the author argues it is more realistic. IMO taking this assumption sidesteps the problem of continual learning rather than solving it. The central problem of continual learning IMO is to learn under non-stationary distribution, the assumption made in this submission makes the distribution more stationary.
3. It is true that this assumption should be utilized when available. However, the only dataset used is manually constructed from CIFAR100, contradicting the initial motivation to move towards a more realistic scenario.
4. There's a lack of information on how the compared methods are adapted to the new scenario.
I searched the supplementary but failed to find a detailed documentation. With the given information, it is hard to tell whether the comparison is fair. My concerns are following,
**increasing from 3 -> 4 as this point is resolved in the rebuttal**
  - In the RandomClasses setting, it is stated that no coreset is used, if the compared methods depends on coreset to replay, it would be unfair. If that's the case, the only conclusion we can draw is that replay is better than no replay, which seems trivial to me.
  - GD depends on internet crawled data, is it replaced with the unlabeled data since it is available in the experiment setting? If not, then I think it is just the setting that favors DistillMatch.
  - With the above said, I suggest the author to list clearly the objectives, replay buffer sizes or even pseudo code for each of the compared method and their own method in a table, which will help the reader identify what major component in the proposed method is making the contribution.

Regarding the quality and clarity,
I found myself confused and making guesses sometimes while reading it.

To list a few:
- introduction paragraph 2, ... to determine which unlabeled data is relevant to the incremental task ..., I guess the incremental task means learning the newly observed data, but then for rehearsal we'll pick the unlabeled data which is from the distribution of past tasks.
- section 1, ... save up to 0.23 stored images per processed image over naive rehearsal (compared to Lee) ..., here seems Lee et al is the naive rehearsal. But then "which only saved 0.08" confuses me, seems to be saying Lee saves 0.08 compared to naive rehearsal.
- section 3, ... where data distributions reflect object class correlations between, and among, the labeled and unlabeled data distributions ... not enough information to infer what "reflect" and "object class correlation" means here.
- section 4, ... Let S_{n-1} denote the score of our OoD detector for valid classes of our pseudo-label model ... what is the "valid classes" needs to be clarified. As I understand it, S_{n-1} measures how likely the unlabeled data is in the distribution of past tasks.
- Super class / Parent class are not defined clear enough.

---

> ### Author Response · Authors · 2020-11-18
> **Authors' response to reviewer #1 (1/2)**
>
> Thank you for the constructive comments. We have replied to your points below.
>
> (1) “The novelty mostly comes from the task scenario, the DistillMatch method is incremental.”
>
> **Response**: We agree that a significant component of our contribution is the proposed setting. From a methodological viewpoint, we contribute the following: we are the first to explore hard knowledge distillation in the continual learning setting (i.e. using one-hot label supervision from a teacher); first to apply Out-of-Distribution (OoD) detection in the continual learning setting (which requires a non-intuitive training and calibration strategy described in Section 4); first to use unlabeled data to balance the classifier in this setting; state-of-the-art performance on all baselines.
>
> (2+3) “Although SSCL is a new scenario, and the author argues it is more realistic. IMO taking this assumption sidesteps the problem of continual learning rather than solving it. The central problem of continual learning IMO is to learn under non-stationary distribution, the assumption made in this submission makes the distribution more stationary… It is true that this assumption should be utilized when available. However, the only dataset used is manually constructed from CIFAR100, contradicting the initial motivation to move towards a more realistic scenario.”
>
> **Response**: Our setting is a realistic realization of the continual learning problem; the motivation for the SSCL method is not to avoid the challenge of continual learning but instead to propose that a slight modification to the problem, SSCL, is more representative of the real-world applications associated with continual learning while remaining challenging. In the real world, where a continual agent’s learning task is a product of its environment, there is an intrinsic correlation between this learning task and the existing unlabeled datastream due to the underlying structure of the environment. We explore the case where the unlabeled data is non-stationary, as Table 2.a explores several realistic non-stationary unlabeled data sequences. We show that prior methods work well only under some of these distributions, and we develop a method that can work across many different types of unlabeled distributions. The aspect of realism that we focus on in the super-class experiments is a task order that is not uniform random, and that as such an unknown correlation exists between the continual learner’s labeled and unlabeled data. In this way we acknowledge that the SSCL setting is not literally the real world, but instead a more realistic continual learning setting that, by design, can be studied with any conventional dataset. We chose to rigorously explore CIFAR-100 because the super-class structure, key to our realistic setting, is well defined, allowing us to design and evaluate a method in the research setting.
>
> (4) “there's a lack of information on how the compared methods are adapted to the new scenario. I searched the supplementary but failed to find a detailed documentation. With the given information, it is hard to tell whether the comparison is fair.”
>
> **Response**: The compared methods are adapted to use unlabeled data for their respective distillation losses. We have highlighted this in the newly added Table 8 (Supplementary I). Note that E2E, DR, and GD all use both coreset (if available) and unlabeled data from the environment, so the comparison is fair.
>
> (5) “In the RandomClasses setting, it is stated that no coreset is used, if the compared methods depends on coreset to replay, it would be unfair. If that's the case, the only conclusion we can draw is that replay is better than no replay, which seems trivial to me.”
>
> **Response**: The compared methods (GD, DR, E2E) leverage unlabeled data for their respective distillation losses, just like our method, so the comparison is fair. The poor performance reflects that the distillation losses for these methods cannot take full advantage of the unlabeled data, indicating the need for our method. We do coreset experiments, and they show that while coresets are important they are not singularly important, as the prior work baselines outperform naive rehearsal, or replay, of the coreset, and our method outperforms naive rehearsal and all prior work baselines.

---

> > ### Author Response · Authors · 2020-11-18
> > **Authors' response to reviewer #1 (2/2)**
> >
> > (6) “GD depends on internet crawled data, is it replaced with the unlabeled data since it is available in the experiment setting? If not, then I think it is just the setting that favors DistillMatch.”
> >
> > **Response**: Yes, it is replaced for a fair comparison. Both our method and GD use the unlabeled dataset, and we show that our method is able to more effectively leverage it.
> >
> > (7) “With the above said, I suggest the author to list clearly the objectives, replay buffer sizes or even pseudo code for each of the compared method and their own method in a table, which will help the reader identify what major component in the proposed method is making the contribution.”
> >
> > **Response**: Thank you for the suggestion. We have added this table to the Supplementary (I) - Table 8. The effects of each component for DM can also be seen in the ablation, Table 2.b.
> >
> > (A) “introduction paragraph 2, ... to determine which unlabeled data is relevant to the incremental task ..., I guess the incremental task means learning the newly observed data, but then for rehearsal we'll pick the unlabeled data which is from the distribution of past tasks.”
> >
> > **Response**: Correct, but the agent does not know which unlabeled data is in the distribution of past tasks.
> >
> > (B) “section 1, ... save up to 0.23 stored images per processed image over naive rehearsal (compared to Lee) ..., here seems Lee et al is the naive rehearsal. But then "which only saved 0.08" confuses me, seems to be saying Lee saves 0.08 compared to naive rehearsal.”
> >
> > **Response**: Your interpretation is correct, and we will edit the paper appropriately to clarify this point. We are saying that our method replaces storing up to 0.23 images per processed unlabeled image, compared to 0.08 images per processed unlabeled image in GD. That is, our method extracts more information from the unlabeled data compared to GD (a large number means the extraction is more efficient).  As shown in Fig. 3, this is equivalent to saying that rehearsal requires 935 stored images to match the performance of our method and 338 images to match the performance of GD.
> >
> > (C) “section 3, ... where data distributions reflect object class correlations between, and among, the labeled and unlabeled data distributions ... not enough information to infer what "reflect" and "object class correlation" means here.”
> >
> > **Response**: This is a high level description stating that the data distributions contain object class correlations. Later in the same section, we describe in lower detail how this is implemented (and visualize the setting in Supplementary F). This is done through the super-class structure in CIFAR100.
> >
> > (D) “section 4, ... Let S_{n-1} denote the score of our OoD detector for valid classes of our pseudo-label model ... what is the "valid classes" needs to be clarified. As I understand it, S_{n-1} measures how likely the unlabeled data is in the distribution of past tasks.”
> >
> > **Response**: These are classes from the previous tasks. Thanks for raising this issue; we have updated our text to clarify this.
> >
> > (E) “Super class / Parent class are not defined clear enough.”
> >
> > **Response**: Our goal is to create a realistic super-class “environment” structure within our SSCL experiments. To do this, we assign two additional labels to each object class, as done in [1]. The first is the parent class, which encompasses 5 object classes that are semantically close. For example, the flowers parent class consists of orchids, poppies, roses, sunflowers, and tulips. We use the parent classes to define our realistic task sequences in Tables 1.b and 2 (there are 20 parent classes total, so the task sequence is 20 tasks). The second additional label is the super-class, which we use to form “environments”. There are eight unbalanced super-classes in which the 20 parent classes are divided into. For example, one super-class contains the parent classes: flowers, fruit/vegetables, and trees. The parent classes and super-classes are only used to form the task sequences, and the learner does not have access to these labels. This is further visualized with examples in Figures 7 and 8 (Supplementary F).
> >
> > [1] Xinqi Zhu and Michael Bain. B-cnn: branch convolutional neural network for hierarchical classifica-tion.arXiv preprint arXiv:1709.09890, 2017.

---

### Official Review · AnonReviewer3 · 2020-10-28
**Recommendation to Accept**

**Rating:** 7
**Confidence:** 4

**Review:**

The paper presents a novel semi-supervised continual learning (SSCL) setting, where labeled data is scarce and unlabeled data is plentiful. The proposed framework is built on pseudo-labeling, consistency regularization, Out-of-Distribution (OoD) detection,
and knowledge distillation in order to reduce the catastrophic forgetting in the proposed setting.

The paper is in general clear and well-written. The contributions are clearly highlighted and the proposed approach is conveniently compared with other state of the art methods, demonstrating its superiority.

Positive aspects:
-  the definition of a realistic, semi-supervised setting for continual learning
- a novel approach for continual learning in order to cope with 'catastrophic forgetting'
- the proposed approach is memory efficient, since it does not need exemplars to replay past tasks

Negative aspects:
- the OoD implemented in this paper rejects the unknown samples. In other words, all unknown samples are considered a single class. It would have been a plus to distinguish between several unknown classes and somehow introduce them in the framework
- the lack of recabilibration step after a number of tasks (in the case of pseudo-labeled samples), could lead to an undesired error propagation which is not quantified in the paper

However, I have some questions:
1. What is the relationship between the 'fi' and 'theta' models (section 4)? Are they completely separate or there is a relationship between them?
For instance, when 'theta' is extended with a new task, is 'fi' extended accordingly? Or is 'fi' trained off-line from the beginning (with all tasks)?
2. There are some different source of errors: distilation, pseudo-labels... Do you perform any kind of system re-calibration? After how many tasks? I mean, do you make a study of error propagation of pseudo-labeled data? Or at some point do you have a human-in-the-loop to correct mis-classification? What is the mis-classification error of pseudo-labeled samples?
3. Do you assume that labeled and unlabeled data come from different distributions or you have a single distribution which is
divided in labeled and unlabeled data at the beginning of the process?
4. Does your scenario foresee that when learning a new task T, all the previous tasks are represented (1..T-1) in the unlabebeld data or only a subpart? (i.e. kind of selective replay)
5. When the number of tasks increases, the number of unlabeled data per task remains constant or is scaled accordingly (i.e. reduced) ?
6. Would be interesting to test your approach in a real-world scenario, i.e. robot navigation.

---

> ### Author Response · Authors · 2020-11-18
> **Authors' response to reviewer #3 (1/2)**
>
> Thank you for the constructive comments and appreciation of our novel setting, approach, and memory efficiency. We have replied to your points below.
>
> (A) “the OoD implemented in this paper rejects the unknown samples. In other words, all unknown samples are considered a single class. It would have been a plus to distinguish between several unknown classes and somehow introduce them in the framework”
>
> **Response**: Thank you for the suggestion. This would require a clustering-like mechanism to separate semantic categories, and maintain them over time, which is not a trivial problem. This is a great idea to explore in follow-up work.
>
> (B) “the lack of recabilibration step after a number of tasks (in the case of pseudo-labeled samples), could lead to an undesired error propagation which is not quantified in the paper”
>
> **Response**: Thanks for pointing this out; we can discuss this in the paper. Because pseudo-labels in task n are generated using theta_{n-1}, then the degradation of pseudo-labels can be quantified with the accuracy/forgetting metrics of theta. We consider this to be part of the continual learning paradigm, and argue that other soft distillation methods have a similar weakness (where the distilling copy is presenting unreliable knowledge). In practice, our method is more robust to this effect (based on the performance) but of course not immune to it. Adding more sophisticated calibration across the tasks is an interesting area of research.
>
> (Q1) “What is the relationship between the 'fi' and 'theta' models (section 4)? Are they completely separate or there is a relationship between them? For instance, when 'theta' is extended with a new task, is 'fi' extended accordingly? Or is 'fi' trained off-line from the beginning (with all tasks)?”
>
> **Response**: These are separate models. We train ‘fi’ with only a subset of the training data so that the OoD detection threshold can be calibrated. The classification accuracy of ‘fi’ would be worse compared to ‘theta’, but this does not matter because ‘fi’ is only used for OoD detection.
>
> (Q2) “There are some different source of errors: distillation, pseudo-labels... Do you perform any kind of system re-calibration? After how many tasks? I mean, do you make a study of error propagation of pseudo-labeled data? Or at some point do you have a human-in-the-loop to correct mis-classification? What is the mis-classification error of pseudo-labeled samples?”
>
> **Response**: We consider this to be a fundamental challenge part of the continual learning paradigm, wherein the forgetting is amplified through task sequences. The other methods have a similar weakness (discussed above in B). Ideally in continual learning, an agent would revisit a task to recalibrate the system, but we do not explore this scenario (and we are not aware of related works which do). However, these are all very interesting proposals for future work.
>
> (Q3) “Do you assume that labeled and unlabeled data come from different distributions or you have a single distribution which is divided in labeled and unlabeled data at the beginning of the process?”
>
> **Response**: We explore both scenarios in Table 2.a. Importantly, our method performs SOTA regardless of whether the distributions are the same, different, or mixed. We show that prior methods are specialized to some distributions over others, in the sense that their performance degrades under some distributions. A key contribution of our method is that we show that our method works well across many distributions.

---

> > ### Author Response · Authors · 2020-11-18
> > **Authors' response to reviewer #3 (2/2)**
> >
> > (Q4) “Does your scenario foresee that when learning a new task T, all the previous tasks are represented (1..T-1) in the unlabebeld data or only a subpart? (i.e. kind of selective replay)”
> >
> > **Response**: In the realistic SSCL case (Table 2), classes represented in the unlabeled data are a subset of classes from the corresponding unlabeled data correlation. Visualized in Figure 8, this correlation can be: (i) positive - unlabeled data is from the same super class, (ii) negative - unlabeled data is from a random different super class, or (iii) random - unlabeled data is a random subset of classes. In the standard SSCL case (Table 1), the classes represented in the unlabeled data are classes across all past, current, and future classes. As such, the unlabeled data may have either all previous tasks or a subpart, or indeed none of the past tasks; it isn’t known and the agent must be robust to all scenarios. This illustrates the need for OoD detection as part of our method.
> >
> > (Q5) “When the number of tasks increases, the number of unlabeled data per task remains constant or is scaled accordingly (i.e. reduced) ?”
> >
> > **Response**: Yes, the number of unlabeled data per task is scaled accordingly.
> >
> > (Q6) “Would be interesting to test your approach in a real-world scenario, i.e. robot navigation.”
> >
> > **Response**: We agree and look forward to exploring this in future work. In this work, we developed a dataset based on CIFAR-100 so that we can perform rigorous experimentation/methodology to be able to singulate the characteristics we are interested in (e.g. exploring different unlabeled distributions). We agree that applying our developed method on unconstrained real robotics data would be very interesting, and is a future direct we are pursuing.

---

### Official Review · AnonReviewer2 · 2020-10-28
**The proposed task is interesting, but more experiments are required**

**Rating:** 6
**Confidence:** 5

**Review:**

- Summary:
This paper proposes class-incremental learning with unlabeled data correlated to labeled data, and a method to tackle it. The task can be considered as a variant of [Lee et al.], which has no assumption on the unlabeled dataset, while this paper assumes the correlation between labeled and unlabeled dataset explicitly. The proposed method is inspired by state-of-the-art class-incremental learning, semi-supervised learning, and out-of-distribution (OoD) detection methods: local distillation [Li and Hoiem], OoD detection [Hsu et al.], consistency regularization and pseudo labeling (or hard distillation) [Sohn et al.], and loss balancing based on class statistics [Lee et al.]. Experimental results support that the proposed method outperforms prior works in the proposed task.

- Reasons for score:
1. Extending continual learning to the semi-supervised setting is natural, given that the extension to self-taught learning has already been considered in [Lee et al.]. However, I cannot agree that semi-supervised learning is more realistic than self-taught learning, which is emphasized throughout the paper 18 times. In an early work of [Raina et al.], self-taught learning is proposed to make the scenario of learning with unlabeled data "widely applicable to many practical learning problems." [Oliver et al.] also argued that "(unlabeled data from out-of-distribution) violates the strict definition of semi-supervised learning, but it nevertheless represents a common use-case for semi-supervised learning (for example, augmenting a face recognition dataset with unlabeled images of people not in the labeled set)." I am not saying that semi-supervised learning is unrealistic, but the argument in this paper sounds overclaimed. I believe both semi-supervised and self-taught learning are realistic in some cases. I also recommend to provide real world scenarios that the proposed task (correlation between labeled and unlabeled data exists and no memory for coreset is available) is useful in practice.
2. The proposed method is not novel, which is essentially the combination of state-of-the-art methods in relevant tasks. But I do not discount this much, because this work would be valuable as the proposed task is interesting but not investigated before. However, the name of task might need to be changed, because a similar name, "semi-supervised incremental learning" is already taken by a kind of semi-supervised learning, which incrementally incorporates unlabeled data to training.
3. Though the improvement over prior class-incremental learning methods is impressive, the overall performance is still too low. In fact, the scale of the experimental setting is too small, so I doubt it is scalable. All experiments are bounded on CIFAR-100, and even only 20% of training data are used as labeled one. Frankly, in this small-scale setting (in both number of data and image resolution), keeping all data is just fine, as the coreset size is negligible compared to the model size. I recommend to experiment in large-scale settings, e.g., on ImageNet. Also, I recommend to compare the oracle setting as well, which keeps all previous training data.
4. In addition to small-scale experimental setting, the architecture is larger than the prior work [Lee et al.]: WRN-28-2 vs. WRN-16-2. In the worst case scenario, it is possible that the best performance of the proposed method is simply from the complexity of their learning objective, i.e., all methods overfit to training data, but the proposed method did not have enough updates to overfit to them.
5. In Figure 3, why do GD and DM not have a coreset? I think there is no reason to give an unfair constraint to them. I recommend to draw curves with respect to increasing number of coreset for those methods as well.
6. Could you provide results on the self-taught learning setting like [Lee et al.]? It would also be interesting to see the performance of the proposed method in the setting.
7. Hyperparameter sweep results provided in Table 4 are either minimum or maximum of the range, so you could improve the performance by enlarging the range.

- Minor Comments:
8. Subscripts of theta often are dropped. Is theta equal to $\theta_{n,1:n}$?
9. "the parameters of no more than three models" -> I believe it is four, because you need to temporarily store gradients during training.
10. $\hat{q}$ is not a probability vector, which makes eq. (2) mathematically do not make sense.
11. Citation format issue: you can use \citet for noun and \citep for adverb.
12. typo on page 5: statoe -> state
13. Table 4: what is TPR here? threshold for consistency regularization?

[Raina et al.] Self-taught Learning: Transfer Learning from Unlabeled Data. In ICML, 2007.

[Li and Hoiem] Learning without forgetting. In TPAMI, 2017.

[Oliver et al.] Realistic Evaluation of Deep Semi-Supervised Learning Algorithms. In NeurIPS, 2018.

[Lee et al.] Overcoming catastrophic forgetting with unlabeled data in the wild. In ICCV, 2019.

[Hsu et al.] Generalized odin: Detecting out-of-distribution image without learning from out-of-distribution data. In CVPR, 2020.

[Sohn et al.] Fixmatch: Simplifying semi-supervised learning with consistency and confidence. In NeurIPS, 2020.

**After rebuttal**

I'd like to thank authors for their efforts to address my concerns. They have addressed most of them, so I increased my score from 5 to 6.

However, there are two concerns that couldn't be resolved during the rebuttal period:

(1) I am still not sure if the proposed task is practical. At glance it looks realistic, but I couldn't find a detailed scenario that can only be solved by the proposed task. Any real world scenario I can think of is closer to [Lee et al.], which is a prior work of this paper. Authors provided an exploring robot example in the thread of responses, but I think [Lee et al.] fits better for the provided one. I recommend authors to find a concrete use-case in real-world applications, which can only be solved by the proposed setting (or at least [Lee et al.] is not applicable; in the revised intro, you may emphasize that there are some real-world problems that [Lee et al.] is not applicable but yours is). R1 and R4 seem to have a similar concern.

(2) the scale of experiment is too small. As CIFAR-10/100 have a limited number of data for your purpose,  you can borrow some data from tinyimages (FYI, CIFAR-10/100 are a subset of 80M tinyimages) or focus on ImageNet.

I am okay with the lack of novelty on the proposed method. For a newly proposed task, I think proposing a simple and effective baseline is good enough. However, because of the two concerns above, I cannot strongly agree with its acceptance.

---

> ### Author Response · Authors · 2020-11-18
> **Authors' response to reviewer #2 (1/2)**
>
> Thank you for the constructive comments. We have replied to your points below.
>
> (1) “Extending continual learning to the semi-supervised setting is natural, given that the extension to self-taught learning has already been considered in [Lee et al.]. However, I cannot agree that semi-supervised learning is more realistic than self-taught learning, which is emphasized throughout the paper 18 times. In an early work of [Raina et al.], self-taught learning is proposed to make the scenario of learning with unlabeled data "widely applicable to many practical learning problems." [Oliver et al.] also argued that "(unlabeled data from out-of-distribution) violates the strict definition of semi-supervised learning, but it nevertheless represents a common use-case for semi-supervised learning (for example, augmenting a face recognition dataset with unlabeled images of people not in the labeled set)." I am not saying that semi-supervised learning is unrealistic, but the argument in this paper sounds overclaimed. I believe both semi-supervised and self-taught learning are realistic in some cases. I also recommend to provide real world scenarios that the proposed task (correlation between labeled and unlabeled data exists and no memory for coreset is available) is useful in practice.”
>
> **Response**: We address the comment on self-taught learning below in (6). Practical examples which reflect our realistic SSCL setting include robotics which continually explore the world and have memory constraints or protected data which cannot be stored for legal reasons. We are not arguing that semi-supervised learning is more realistic than self-taught learning, this is a mistake of terminology and we apologize for the confusion. Like Lee et. al. we are closer in spirit to self-taught learning as each task’s unlabeled data is not drawn from the same distribution as the task’s label data.
>
> (2) “The proposed method is not novel, which is essentially the combination of state-of-the-art methods in relevant tasks. But I do not discount this much, because this work would be valuable as the proposed task is interesting but not investigated before. However, the name of task might need to be changed, because a similar name, "semi-supervised incremental learning" is already taken by a kind of semi-supervised learning, which incrementally incorporates unlabeled data to training.”
>
> **Response**: We agree that a significant component of our contribution is the proposed setting.  From a methodological viewpoint, we contribute the following: we are the first to explore hard knowledge distillation in the continual learning setting (i.e. using one-hot label supervision from a teacher); first to apply Out-of-Distribution (OoD) detection in the continual learning setting (which requires a non-intuitive training and calibration strategy described in Section 4); first to use unlabeled data to balance the classifier in this setting; state-of-the-art performance on all baselines. Thank you for the feedback on the name of the task, we will consider this.
>
> (3) “Though the improvement over prior class-incremental learning methods is impressive, the overall performance is still too low. In fact, the scale of the experimental setting is too small, so I doubt it is scalable. All experiments are bounded on CIFAR-100, and even only 20% of training data are used as labeled one. Frankly, in this small-scale setting (in both number of data and image resolution), keeping all data is just fine, as the coreset size is negligible compared to the model size. I recommend to experiment in large-scale settings, e.g., on ImageNet. Also, I recommend to compare the oracle setting as well, which keeps all previous training data.”
>
> **Response**: We use the Omega metric to report performance with respect to the upper bound (i.e. oracle setting), and have added the oracle performance to the results tables, thank you for the suggestion. The oracle final accuracy is 56.7%. We chose the CIFAR-100 dataset due to the well defined super-class structure, which we consider the most important aspect of our experiment section. It allows us to rigorously explore our underlying setting and research question (i.e. with and without class correlations in the unlabeled set, etc.). Moreover CIFAR-100 is a staple both in the semi-supervised and continual learning communities.

---

> > ### Author Response · Authors · 2020-11-18
> > **Authors' response to reviewer #2 (2/2)**
> >
> > (4) “In addition to small-scale experimental setting, the architecture is larger than the prior work [Lee et al.]: WRN-28-2 vs. WRN-16-2. In the worst case scenario, it is possible that the best performance of the proposed method is simply from the complexity of their learning objective, i.e., all methods overfit to training data, but the proposed method did not have enough updates to overfit to them.”
> >
> > **Response**: We believe the comparison is fair: We re-implemented the prior work in our setting and evaluate both using the WRN-28-2 architecture and the same number of updates. Similar to the prior work, we also tuned the hyper-parameters with validation data to avoid overfitting to the training data.
> >
> > Using the WRN-16-2 architecture, we have replicated the ten-task experiment from Table 1 (CIFAR-100 with 20% Labeled Data for RandomClass Tasks, 10 tasks, no Coreset, Uniform Unlabeled Data Distribution) and see that the results are very similar to WRN-28-2 (with our method performing best).
> >
> > --------------------------------------------\
> > &emsp;&nbsp;&nbsp; Method &emsp;&nbsp;| A_n | Omega \
> > --------------------------------------------\
> > Upper Bound | 56.5 | 100.0 \
> > --------------------------------------------\
> > &emsp;&emsp; Base &emsp;&emsp;&nbsp;| &nbsp;8.2&nbsp; | 34.4 \
> > &nbsp;&emsp;&emsp; E2E &emsp;&emsp;&nbsp;&nbsp;| &nbsp;7.6 &nbsp;| 32.1 \
> > &nbsp;&nbsp;&emsp;&emsp; DR &emsp;&emsp;&nbsp;&nbsp;| &nbsp;8.2&nbsp; | 34.8 \
> > &nbsp;&nbsp;&emsp;&emsp; GD &emsp;&emsp;&nbsp;&nbsp;| 23.1 | 63.1 \
> > --------------------------------------------\
> > &emsp;DM (ours)&emsp; | **38.1** | **79.6** \
> > --------------------------------------------\
> > (5) “In Figure 3, why do GD and DM not have a coreset? I think there is no reason to give an unfair constraint to them. I recommend to draw curves with respect to increasing number of coreset for those methods as well.”
> >
> > **Response**: The purpose of Fig. 3 is to show how many stored images are required by the baseline method to match the performance of DM and GD with no stored images. Thus, we show these methods with horizontal lines and solve for the intersections. Moreover, for some of the experiments we do use a coreset for our method DM (Tables 1.b and 2), as well as GD and the other baselines.
> >
> > (6) “Could you provide results on the self-taught learning setting like [Lee et al.]? It would also be interesting to see the performance of the proposed method in the setting.”
> >
> > **Response**: Thank you for the suggestion - we think the self-taught learning setting would be interesting from a continual learning perspective but is distinctly different from our setting. Our understanding of self-taught learning is that representations are learned using only the unlabeled data, and then the labeled data is embedded into this representation space. While it would be interesting to investigate the task of learning unsupervised representations using unlabeled data, we think this is a different problem that requires a mechanism to learn unsupervised representations (which is orthogonal to our problem setting and method).This is an interesting topic of future work.
> >
> > (7) “Hyperparameter sweep results provided in Table 4 are either minimum or maximum of the range, so you could improve the performance by enlarging the range.”
> >
> > **Response**: Thank you for this comment. This was a typo and we have updated the supplementary material to reflect this.
> >
> > (8) “Subscripts of theta often are dropped. Is theta equal to $\theta_{n,1:n}$?”
> >
> > **Response**: Yes, and thank you for this comment. We have fixed this in the paper.
> >
> > (9) “"the parameters of no more than three models" -> I believe it is four, because you need to temporarily store gradients during training.”
> >
> > **Response**: This is correct. Our intent is to clarify that the model parameters do not exceed that of GD, and we have revised the paper to clarify this point.
> >
> > (10) “\hat_q is not a probability vector, which makes eq. (2) mathematically do not make sense”
> >
> > **Response**: Thank you for pointing this out. This should be $q$; the paper is updated with this fixed.
> >
> > (11-12)
> >
> > **Response**: Thank you for pointing out these errors.
> >
> > (13) “Table 4: what is TPR here? threshold for consistency regularization?”
> >
> > **Response**: Defined in Section 4: OoD Detector Training and Calibration, TPR is the True-Positive-Rate at which the OoD detector is calibrated for.

---

> > > ### Comment · AnonReviewer2 · 2020-11-23
> > > **Unresolved issues and minor comments**
> > >
> > > Unresolved issues and minor comments
> > >
> > > Thanks for your responses. I am either satisfied or okay with most of the responses, except the followings:
> > >
> > > (3) I think your response does not answer the question "only 20% of training data (on CIFAR-100) are used as labeled one. Frankly, in this small-scale setting (in both number of data and image resolution), keeping all data is just fine, as the coreset size is negligible compared to the model size."
> > >
> > > I agree that CIFAR-100 has been commonly used as a benchmark dataset in continual learning. However, your experimental setting has only 20% of training data. I am not sure if this is scalable, such that the presented results are consistent in real world applications. This also aligns with the concern from R1.
> > > Compared to GD, while GD's setting has 50k training data and 1M unlabeled data retrieved per stage, your setting has 10k training data and 40k unlabeled data. Another reason why I asked to try the same experimental setting with GD in (6) is to partially avoid the scalability issue.
> > >
> > > (R3,Q5) When the number of tasks increases, the number of unlabeled data per task is scaled accordingly.
> > >
> > > I understand why you had to do this, maybe because you don't want to have unlabeled data from unknown labels at each task. But is this a realistic setting? For example, if your model is learning to recognize animal species continually, e.g., dog, cat, and bear species in sequence, how do you ensure that only unlabeled dogs are appeared when training dogs, only unlabeled dogs and cats are appeared when training cats (only dogs should additionally be appeared among non-cats), and so on? If you filter out unknown animals, it becomes the setting in GD, and unlabeled cats should be filtered out as well, because they are not trained yet.
> > >
> > > This simple continual classifier learning scenario would not fit to your target scenario, so I asked to provide real world scenarios that needs the proposed setting in (1). Your answer, "Practical examples which reflect our realistic SSCL setting include robotics which continually explore the world" is not that specific. I think the setting in GD (no assumption between labeled and unlabeled data) can be proposed with the same statement.
> > >
> > > (R1, Cons 4-2) though it is not so relevant to R1's concern, how did you treat the sampling method of GD? Did you simply ignore their sampling method? Or, if you sampled unlabeled data further, how many unlabeled data are sampled?

---

> > > > ### Author Response · Authors · 2020-11-24
> > > > **Authors' response to reviewer #2 unresolved issues (2/2)**
> > > >
> > > > “This simple continual classifier learning scenario would not fit to your target scenario, so I asked to provide real world scenarios that needs the proposed setting in (1). Your answer, ‘Practical examples which reflect our realistic SSCL setting include robotics which continually explore the world’ is not that specific. I think the setting in GD (no assumption between labeled and unlabeled data) can be proposed with the same statement.”
> > > >
> > > > **Response**: Thank you for this comment, and we agree that our example should be more specific. As shown in Figure 1, the difference between our setting and the GD setting is that our setting assumes no access to an external datastream (such as web data) and instead relies on the environment for replay. This is why we exemplify this with an exploring robot: an agent exploring an area with no access to communication, like the jungle, space, or the sea, would not have access to a large external datastream. However, it could look to its unlabeled environment data for instances of known objects for replay. In the space example, a planetary rover tasked with classifying 10 types of materials might then be given examples of 5 more types of materials to learn. To update its model in situ, this robot could augment its new labeled data with the data from its surrounding, which may contain material it can already classify, examples of the new materials, or materials in neither group, and with DistillMatch be able to appropriately distinguish between them. In summary, the key question in our paper is whether unlabeled data from the environment itself can improve continual learning performance.
> > > >
> > > > (R1, Cons 4-2) though it is not so relevant to R1's concern, how did you treat the sampling method of GD? Did you simply ignore their sampling method? Or, if you sampled unlabeled data further, how many unlabeled data are sampled?
> > > >
> > > > **Response**: Given that all unlabeled data in our setting can be used by GD (i.e., sampling is not necessary due to resource budget issues), and the unlabeled data already contains a mixture of in-domain and out-of-domain data (as sampled for in GD), we use all available unlabeled data from the environment in our implementation of GD.  We note that when developing our method, we found that unlabeled data can only hurt performance when applying a hard distillation loss (i.e., calculating a pseudo-label and performing standard cross entropy loss). Therefore, we expect a similar trend for GD (i.e., that reducing the amount of unlabeled data used by GD with sampling would either hurt performance or have no effect, but not help performance). We believe reducing the amount of unlabeled data for GD would make it perform closer to the non-unlabeled data baseline, and will run this experiment for the final paper.

---

> > > > ### Author Response · Authors · 2020-11-24
> > > > **Authors' response to reviewer #2 unresolved issues (1/2)**
> > > >
> > > > Thank you for the response and follow-up comments. We have replied to your remaining points below.
> > > >
> > > > (3) “I think your response does not answer the question ‘only 20% of training data (on CIFAR-100) are used as labeled one. Frankly, in this small-scale setting (in both number of data and image resolution), keeping all data is just fine, as the coreset size is negligible compared to the model size.’ I agree that CIFAR-100 has been commonly used as a benchmark dataset in continual learning. However, your experimental setting has only 20% of training data. I am not sure if this is scalable, such that the presented results are consistent in real world applications. This also aligns with the concern from R1. Compared to GD, while GD's setting has 50k training data and 1M unlabeled data retrieved per stage, your setting has 10k training data and 40k unlabeled data. Another reason why I asked to try the same experimental setting with GD in (6) is to partially avoid the scalability issue.”
> > > >
> > > > **Response**: To help address the scalability concerns, we have added a supplementary section (J) which compares our method and the competing methods using the Tiny-ImageNet dataset (200 classes of 64x64 resolution images with 500 training images per class). We experimented in a similar setting to Table 1.a (20% labeled data with RandomClass Tasks, no Coreset, Uniform Unlabeled Data Distribution) with a ten-task sequence (20 classes per task). In this experiment, there are 20,000 labeled images and 80,000 unlabeled images; thus, we both double the number of data and double the image resolution. We find that the conclusions from Table 1.a in our paper scale to this experiment.
> > > >
> > > > --------------------------------------------\
> > > > &emsp;&nbsp;&nbsp; Method &emsp;&nbsp;| A_n | Omega \
> > > > --------------------------------------------\
> > > > Upper Bound | 40.7 | 100 \
> > > > --------------------------------------------\
> > > > &emsp;&emsp; Base &emsp;&emsp;&nbsp;| &nbsp;6.5&nbsp; | 35.1 \
> > > > &nbsp;&emsp;&emsp; E2E &emsp;&emsp;&nbsp;&nbsp;| &nbsp;5.8 &nbsp;| 30.3 \
> > > > &nbsp;&nbsp;&emsp;&emsp; DR &emsp;&emsp;&nbsp;&nbsp;| &nbsp;6.8&nbsp; | 35.3 \
> > > > &nbsp;&nbsp;&emsp;&emsp; GD &emsp;&emsp;&nbsp;&nbsp;| 11.9 | 50.6 \
> > > > --------------------------------------------\
> > > > &emsp;DM (ours)&emsp; | **24.8** | **74.7** \
> > > > --------------------------------------------\
> > > > (R3,Q5) “When the number of tasks increases, the number of unlabeled data per task is scaled accordingly. I understand why you had to do this, maybe because you don't want to have unlabeled data from unknown labels at each task. But is this a realistic setting? For example, if your model is learning to recognize animal species continually, e.g., dog, cat, and bear species in sequence, how do you ensure that only unlabeled dogs are appeared when training dogs, only unlabeled dogs and cats are appeared when training cats (only dogs should additionally be appeared among non-cats), and so on? If you filter out unknown animals, it becomes the setting in GD, and unlabeled cats should be filtered out as well, because they are not trained yet.
> > > >
> > > > **Response**: Thank you for this comment. We interpreted R3’s question to be asking whether the number of unlabeled data per task is reduced for longer task sequences to avoid having the same unlabeled examples appear in more than one task. 80% of the CIFAR-100 training data is used as unlabeled data (40,000 images), so we divide this unlabeled data evenly amongst the tasks to avoid images appearing in more than one task. For example, there are 8,000 unlabeled images per class in our five-task sequence experiments and 4,000 unlabeled images per class in our ten-task sequence experiments. We do not prevent unlabeled data from unknown classes from appearing in tasks before they appear in the labeled data; in fact, having unlabeled data from future classes is part of what makes our setting challenging (and why our approach includes OoD detection). In other words, additional classes (beyond the current and past tasks) can actually hurt performance without OoD detection; we believe this is because they will be falsely assigned a pseudo-label for replay, which can interfere with learning the class from labeled data in a later task.

---

### Official Review · AnonReviewer4 · 2020-10-30
**The proposed DistillMatch method appears to be a combination of knowledge distillation, out of distribution detection, consistency regularization and several other small tricks.**

**Rating:** 5
**Confidence:** 3

**Review:**

This paper investigates a semi-supervised continual learning (SSCL) setting and proposes a new method called DistillMatch for this setting. The major contributions are: (1) The authors carefully design a realistic SSCL setting where object-object correlations between labeled and unlabeled sets are maintained through a label super-class structure. And then, they develop the DistillMatch method combining knowledge distillation, pseudo-labels, out of distribution detection, and consistency regularization. (2) They show that  DistillMatch outperforms other existing methods on CIFAR-100 dataset, and ablation study results are shown also.

However, there are some downsides that should be considered before its publication. (1) In abstract the authors claim that they can significantly reduce the memory budget (of labeled training data) by leveraging unlabeled data (perhaps with large volume). This motivation seems to be contradictive. (2) From a methodological viewpoint, the proposed DistillMatch method is just a combination of existing methods (listed as in above). So where is the novelty of this "new" method? (3) In experiments, the chosen baseline algorithm is very weak. There are some strong baseline methods such as GEM, A-GEM, and ER. So I wonder to know the real improvements over state-of-the-art methods for continual learning. (4) The label super-class structure existed in CIFAR-100 has been used in their experiments. But this is not very common for other more realistic datasets such as miniImageNet. If there is no super-class structure, we don't know how to apply the proposed DistillMatch method.

In summary, I think this semi-supervised continual learning setting is interesting, but the proposed DistillMatch method can not persuade me that this method is a novel significant contribution to this problem. So at present time I believe there is much room for the authors to improve their method before publication.

---

> ### Author Response · Authors · 2020-11-18
> **Authors' response to reviewer #4 (1/2)**
>
> Thank you for the constructive comments and interest in our setting.  We have replied to your points below.
>
> (1) “In abstract the authors claim that they can significantly reduce the memory budget (of labeled training data) by leveraging unlabeled data (perhaps with large volume). This motivation seems to be contradictive.”
>
> **Response**: Our claim is that we significantly reduce the memory budget (and not the computational budget) by leveraging unlabeled data existing in the continual learning agent’s environment (i.e. this unlabeled data is never stored, and never needs to be). The assumption we make is that the agent is operating in an “environment” containing both labeled and unlabeled data, not unlike a room in a house with some objects labeled, and unlabeled data is observed, can be used, and discarded, hence incurring no no memory cost; this is highlighted in Fig. 1). While this assumption does not hold for all continual learning applications, it does hold for many important applications such as robots/agents exploring and continuously learning in an environment, and situations in which data is legally sensitive and cannot be stored ( like internet content taken down by content creators).
>
> (2) “From a methodological viewpoint, the proposed DistillMatch method is just a combination of existing methods (listed as in above). So where is the novelty of this "new" method?”
>
> **Response**: We contribute the following: we are the first to explore hard knowledge distillation in the continual learning setting (i.e. using one-hot label supervision from a teacher); first to apply Out-of-Distribution (OoD) detection in the continual learning setting (which requires a non-intuitive training and calibration strategy described in Section 4); first to use unlabeled data to balance the classifier in this setting; state-of-the-art performance on all baselines.
>
> (3) “In experiments, the chosen baseline algorithm is very weak. There are some strong baseline methods such as GEM, A-GEM, and ER. So I wonder to know the real improvements over state-of-the-art methods for continual learning.”
>
> **Response**: We choose to compare to methods which best fit in our setting, which are algorithms that can mitigate catastrophic forgetting with and without a coreset (also called “replay”) and can leverage unlabeled data for fair comparisons. GD, E2E, and DR can all leverage the unlabeled data in their respective distillation losses, and GD was state-of-the-art for continual learning methods that leverage unlabeled data prior to our work. If by ER the reviewer is referring to naive rehearsal, sometimes called “experience rehearsal” or “experience replay”, we do include results for naive rehearsal, listed as “Base” in the experiments with a coreset.  If the author is referring to the CLEAR algorithm from [1], sometimes referred to as ER, this algorithm targets the reinforcement learning domain, which is not considered in our work.
>
> Replay-focused methods such as GEM and A-GEM cannot work in experiments where no coreset is present because they rely on replay. Moreover, they have no mechanism to learn from the unlabeled data; as such it would be an unfair comparison to GEM and A-GEM if we outperformed them using unlabeled data when they could not use any, so the comparison has no null hypothesis. Additionally, the unique contributions of replay-focused methods like GEM and A-GEM are orthogonal to our contribution and can be combined with our approach (and the competing approaches) for better performance. We have added a brief acknowledgement of these details in Section 5.

---

> > ### Author Response · Authors · 2020-11-18
> > **Authors' response to reviewer #4 (2/2)**
> >
> > (4) The label super-class structure existed in CIFAR-100 has been used in their experiments. But this is not very common for other more realistic datasets such as miniImageNet. If there is no super-class structure, we don't know how to apply the proposed DistillMatch method.
> >
> > **Response**: The super-class structure is not known or used during training and is only used to set up the problem setting to simulate different unlabeled distributions. The aspect of realism that we focus on in the super-class experiments is an unlabeled distribution that comes from related objects within a super-class rather than uniformly chosen from all classes. For example, consider a robot learning about objects in the world while exploring that world. More photorealistic or diverse datasets still sampled uniformly; for example, ImageNet, iNaturalist, Places, or Open Images do not test this aspect of realism. Specifically, the super-class structure is used to design the non-uniform continual learning curriculum and tests generalization performance with correlated continual learning curricula. Per our description of DistillMatch in Section 4, the only task information used by DistillMatch is knowledge of when the task changes, and as such has no knowledge of the task order, super-class or otherwise. We chose to rigorously explore CIFAR-100 because this structure is well defined, allowing us to design and evaluate a method in the research setting, and it is heavily used by prior work in semi-supervised learning and continual learning. In summary, the existence of the super-class structure allowed us to rigorously explore and ablate the characteristics of the problem we are studying, but was not used by the methods at all.
> >
> > [1] Rolnick, D., Ahuja, A., Schwarz, J., Lillicrap, T., & Wayne, G. (2019). Experience replay for continual learning. In Advances in Neural Information Processing Systems (pp. 350-360).
> > [2] http://image-net.org/explore

---

### Author Response · Authors · 2020-11-24
**Two key highlights of the authors' responses**

We thank the reviewers again for their constructive comments and feedback. We would like to highlight two key revisions which were made in direct responses to reviewer comments (in addition to our responses to each reviewer).

First, to help address the scalability concerns, we have added a supplementary section (J) which compares our method and the competing methods using the Tiny-ImageNet dataset (200 classes of 64x64 resolution images with 500 training images per class). We experimented in a similar setting to Table 1.a (20% labeled data with RandomClass Tasks, no Coreset, Uniform Unlabeled Data Distribution) with a ten-task sequence (20 classes per task). In this experiment, there are 20,000 labeled images and 80,000 unlabeled images; thus, we both double the number of data and double the image resolution. We find that the conclusions from Table 1.a in our paper scale to this experiment. The results are copied here.

--------------------------------------------\
&emsp;&nbsp;&nbsp; Method &emsp;&nbsp;| A_n | Omega \
--------------------------------------------\
Upper Bound | 40.7 | 100 \
--------------------------------------------\
&emsp;&emsp; Base &emsp;&emsp;&nbsp;| &nbsp;6.5&nbsp; | 35.1 \
&nbsp;&emsp;&emsp; E2E &emsp;&emsp;&nbsp;&nbsp;| &nbsp;5.8 &nbsp;| 30.3 \
&nbsp;&nbsp;&emsp;&emsp; DR &emsp;&emsp;&nbsp;&nbsp;| &nbsp;6.8&nbsp; | 35.3 \
&nbsp;&nbsp;&emsp;&emsp; GD &emsp;&emsp;&nbsp;&nbsp;| 11.9 | 50.6 \
--------------------------------------------\
&emsp;DM (ours)&emsp; | **24.8** | **74.7** \
--------------------------------------------\
Second, to highlight how the compared methods are adapted to use unlabeled data for their respective distillation losses, we have added Table 8 (Supplementary I) which is copied here as well. In this table, “L” refers to labeled training data from the current task, “C” refers to labeled coreset data from past tasks (if available), and “U” refers to unlabeled data from the environment. Note that E2E, DR, and GD all use both coreset (if available) and unlabeled data from the environment, so the comparison is fair.

-----------------------------------------------------------------------------------------------------\
Component  &emsp;&emsp; &emsp;&emsp;&emsp; &emsp;&emsp;&emsp; &emsp; &emsp; Base | E2E&nbsp; | &nbsp;DR&nbsp; | &nbsp;GD&nbsp; | DM (ours) \
-----------------------------------------------------------------------------------------------------\
Classification Loss  &emsp;&emsp; &emsp;&emsp;&emsp; &emsp; &nbsp;&nbsp;&nbsp;&nbsp;L/C &nbsp;|   &nbsp;L/C&nbsp; |  &nbsp; L/C |  &nbsp;L/C&nbsp; |   &nbsp;L/C \
\
Per-Task Distillation &emsp;&emsp; &emsp;&emsp;&emsp; &emsp;   &emsp;  --&nbsp; |   &nbsp; C/U |      &nbsp;&nbsp; --&nbsp;&nbsp;     |   &nbsp;&nbsp; --&nbsp; &nbsp;   |  C/U \
(over Previous Tasks) \
\
Single-Task Distillation   &emsp;&emsp; &emsp;&emsp;&emsp;    &nbsp; --&nbsp; &nbsp; |    &nbsp;&nbsp; --&nbsp; &nbsp;   | &nbsp;  C/U |   C/U |  &nbsp;&nbsp; --&nbsp; &nbsp;  \
(over Previous Tasks) \
\
Distill Current Task &emsp;&emsp; &emsp;&emsp;&emsp; &emsp; &emsp;     &nbsp; --&nbsp; &nbsp; |    &nbsp;&nbsp; --&nbsp;    | &nbsp;  L/U |   L/U &nbsp;|  &nbsp;&nbsp; --&nbsp; &nbsp;  \
(from Separate Trained Model)  \
\
Soft Global Distillation  &emsp;&emsp; &emsp;&emsp; &emsp; &emsp;    --  &nbsp; |    &nbsp;&nbsp; --&nbsp; &nbsp;   | &nbsp;&nbsp; --&nbsp; &nbsp;  |  &nbsp;U &nbsp;&nbsp;|  &nbsp;&nbsp; --&nbsp; &nbsp;  \
(All-Tasks)  \
\
Hard Global Distillation &emsp;&emsp; &emsp;&emsp; &emsp; &nbsp;&nbsp;    --  &nbsp; |    &nbsp; --&nbsp; &nbsp;   | &nbsp;&nbsp; --&nbsp; &nbsp;  |  &nbsp;&nbsp; -- &nbsp;|  &nbsp;U &nbsp;&nbsp;  \
(All-Tasks) \
\
Consistency Loss &emsp;&emsp; &emsp;&emsp; &emsp; &nbsp; &emsp; &emsp;  &emsp;   --  &nbsp; |    &nbsp; --&nbsp; &nbsp;   | &nbsp;&nbsp; --&nbsp; &nbsp;  |  &nbsp;&nbsp; -- &nbsp;|  &nbsp;U &nbsp;&nbsp;  \
(All-Tasks) \
\
Confidence Calibration &emsp;&emsp; &emsp;&emsp; &emsp; &emsp;    --  &nbsp; |    &nbsp;&nbsp; --&nbsp; &nbsp;   | &nbsp;&nbsp; --&nbsp;  &nbsp;|  C/U |  &nbsp;&nbsp; --&nbsp; &nbsp;  \
\
OoD Detection &emsp;&emsp; &emsp;&emsp; &emsp; &nbsp; &emsp; &emsp;  &emsp; &emsp;  --  &nbsp; |    &nbsp; --&nbsp; &nbsp;   | &nbsp;&nbsp; --&nbsp; &nbsp;  |  &nbsp;&nbsp; -- &nbsp;|  &nbsp;U &nbsp;&nbsp;  \
\
Fine-Tuning  &emsp;&emsp;&emsp;&emsp;&emsp;&emsp;&emsp; &emsp;&emsp;&emsp; &emsp; &nbsp;&nbsp;&nbsp;&nbsp; -- &nbsp;&nbsp;&nbsp;|   &nbsp;L/C&nbsp; |  &nbsp; -- &nbsp;|  &nbsp;L/C&nbsp; |   &nbsp;L/C \
\
Class-Balancing &emsp;&emsp;&emsp;&emsp;&emsp; &emsp;&emsp;&emsp; &emsp; &nbsp;&nbsp;&nbsp;&nbsp; -- &nbsp;&nbsp;&nbsp;|   &nbsp;L/C&nbsp; |  &nbsp; -- &nbsp;|  &nbsp;L/C&nbsp; |   &nbsp;L/C/U \
-----------------------------------------------------------------------------------------------------\
 &nbsp;

---

### Decision · Program_Chairs · 2021-01-07
**Final Decision**

**Decision:**

Reject

**Comment:**

This paper proposes a semi-supervised setting to reduce memory budget in replay-based continual learning.
It uses unlabeled data in the environment for replaying which requires no storage, and generates pseudo-labels where unlabeled data is connected to labeled one.
The method was validated on the proposed tasks.

Pros:
- The semi-supervised continual learning setting is novel and interesting.
- The proposed approach is memory efficient, since it does not need exemplars to replay past tasks.

Cons:
- The scale of experiment is small. It lacks evaluation in real world environment.
- The novelty is limited, because it is a combination of existing technologies: pseudo-labeling, consistency regularization, Out-of-Distribution (OoD) detection, and knowledge distillation.
- The comparison might not be fair due to different settings.

The authors addressed the fairness and scalability with additional experiments
and leave some suggestions of reviewers for future work.
R3 had a concern on the error propagation of pseudo-labels which I also share. The authors agreed that this is a challenge for all CL methods.

In summary, the reviews are mixed. All reviewers agree that the semi-supervised continual learning setting is novel and interesting, and some have concerns on scalability and novelty of the method which I also share. So at present time I believe there is much room for the authors to improve their method and experiments before publication.